

# Experimental evidence of the potential bioavailability for marine heterotrophic bacteria of aerosols organic matter

Kahina Djaoudi[1,2], France Van Wambeke[1], Aude Barani[1], Nagib Bhairy[1], Servanne Chevaillier[5],

5    Karine Desboeufs[5], Sandra Nunige[1], Mohamed Labiadh[3], Thierry Henry des Tureaux[4], Dominique Lefèvre[1], Amel Nouara[1], Christos Panagiotopoulos[1], Marc Tedetti[1], Elvira Pulido-Villena[1].

[1] Aix-Marseille Univ., Université de Toulon, CNRS, IRD, MIO UM 110, 13288, Marseille, France

[2] Molecular and Cellular Biology, The University of Arizona, Tucson, USA

[3] IRA (Institut des Régions Arides) de Médenine, El Fjé4119, Tunisia

10   [4] iEES Paris (Institut d'Ecologie et des Sciences de l'Environnement de Paris), UMR IRD 242, Université Paris Est Créteil—Sorbonne Université—CNRS—INRA—Université de Paris, F-93143 Bondy, France

[5] LISA, UMR7583, Université de Paris, Université Paris-Est-Créteil, Institut Pierre Simon Laplace (IPSL), Créteil, France.

*Correspondence to*: kdjaoudi@email.arizona.edu

25





**Abstract.** The surface ocean receives important amounts of organic carbon from atmospheric deposition. The degree of bioavailability of this source of organic carbon will determine its impact on the marine carbon cycle. In this study, the potential availability of dissolved organic carbon (DOC) leached from both desert dust and anthropogenic aerosols to marine heterotrophic bacteria was investigated. The experimental design was based on 16-days incubation, in the dark, of a marine bacterial inoculum into artificial seawater amended with water-soluble Saharan dust (D-treatment) and anthropogenic (A-treatment) aerosols, so that the initial DOC concentration leachate from aerosols is 36 µM C. Glucose-amended (G) and non-amended (control) treatments were run in parallel. Over the incubation period, an increase in bacterial abundance (BA) and bacterial production (BP) was observed first in the G-treatment, followed then by D and finally A treatments, with bacterial growth rates significantly higher in the G and D treatments than the A treatment. Following this growth, maxima of BP reached were similar in D ($879 \pm 64$ ng C $L^{-1}$ $h^{-1}$; n=3) and G ($648 \pm 156$ ng C $L^{-1}$ $h^{-1}$; n=3) treatments and were significantly higher than in A-treatment ($124 \pm 39$ ng C $L^{-1}$ $h^{-1}$; n=2). The DOC consumed over the incubation period was similar in A ($9 \pm 4$ µM; n=2) and D ($9 \pm 2$ µM; n=3) treatments and was significantly lower than that consumed in the G-treatment ($22 \pm 3$ µM). Nevertheless, the bacterial growth efficiency (BGE) in the D treatment ($14.2 \pm 5.5\%$; n=3) compared well with the G treatment ($7.6 \pm 2\%$; n=3), suggesting that the metabolic use of the labile DOC fraction in both conditions was energetically equivalent. In contrast, the BGE in the A- treatment was lower ($1.7 \pm 0.1$ %; n=2), suggesting that the most part of used labile DOC was catabolized. The results obtained in this study highlight the potential of aerosol organic matter to sustain the metabolism of marine heterotrophs and stress the need to include this external source of organic carbon into biogeochemical models, for a better constraining of the carbon budget.

**Key words:** atmospheric deposition; organic carbon; desert dust; anthropogenic aerosols; bioavailability.



## 1. Introduction

Marine dissolved organic matter (DOM) is the largest reservoir of reduced carbon in the ocean. Estimated at 662 Pg C, which is comparable to that present as atmospheric $CO_2$, DOM plays a key role in the ocean carbon cycle as it is an important pathway of carbon export (Hansell et al., 2009; Moran et al., 2016). At the global scale, dissolved organic carbon (DOC) export from the surface to the deep ocean contributes to 20% of the total organic carbon flux (Hansell et al., 2009). This percentage reaches more than 50% of the total carbon export in the oligotrophic oceans (Carlson et al., 1994; Guyennon et al., 2015; Letscher and Moore, 2015; Roschan and DeVeries, 2017).

Nutrient availability and microbial community structure regulate the accumulation and the remineralization of DOM, influencing thus the DOC export efficiency (Carlson et al., 2002; Letscher and Moore., 2015; Romera-Castillo et al., 2016). From extensive field studies, it is well known that DOM flux into heterotrophic bacteria is a major pathway in the regulation of carbon fluxes in the ocean (i.e. Azam et al., 1983; Moran et al., 2016). Heterotrophic bacteria are an especially important component of marine oligotrophic regions in which their biomass is comparable to that of phytoplankton. In such oligotrophic areas, half of oceanic primary production is channeled via heterotrophic bacteria to the microbial loop (Fuhrman, 1992; Azam, 1998), driving a wide range of biogeochemical processes that are important for the carbon cycle (Bunse and Pinhassi, 2017; Gasol and Kirchman, 2018 and references therein). Among these processes, the microbial activity has been identified as involved in the alteration of the chemical composition of the DOM pool, influencing thereby the residence time of the carbon in the ocean (Microbial carbon pump, Jiao et al., 2010).

The open ocean receives from the atmosphere a continuous flux of anthropogenic particles, resulting from both industrial and agricultural activities, as well as pulsed fluxes of natural origin such as desert dust (De Leeuw et al., 2014 and references therein). During their transport to the ocean, dust particles mix with anthropogenic aerosols/gas supplying the water column with a wide variety of compounds including macro- and micro-nutrients (N, P, Fe…) (Duce et al., 1991; Jickells et al., 2005), as well as potentially toxic elements (Paytan et al., 2009; Jordi et al., 2012). By bringing new nutrients to the upper waters, atmospheric deposition plays a key role in the stratified oligotrophic regions (Guieu et al., 2014). The relative response of phytoplankton and heterotrophic bacteria to dust deposition has been shown to depend on the nutritional status of the environment in which they develop. Indeed, the reported positive effect of dust deposition on the primary production in the central Atlantic Ocean decreased with increasing oligotrophy of the seawater (Maranon et al., 2010), suggesting a competitive advantage of heterotrophic bacteria over phytoplankton in the oligotrophic ocean. Nevertheless, recently, the contrasted influence of Saharan dust *versus* anthropogenic aerosols on bacterioplankton composition and metabolism is getting attention (Herut et al., 2016; Marín et al., 2017a).

Most studies on the biogeochemical role of atmospheric deposition have focused on the potential of inorganic compounds to relieve nutrient limitation (i.e. Duce et al., 1991; Guieu et al., 2014). However, there is increasing evidence that a significant fraction of atmospheric deposition occurs as organic forms (Duce et al., 2008; Kanakidou et al., 2012; Djaoudi et al., 2018; Violaki et al., 2018; Vila-Costa et al., 2019). The extent of organic compounds coating onto dust has been related, among others, to the transport pathway and the reactivity of organic species in the



atmosphere (Falkovich et al., 2004; Dall'Osto et al., 2010; Theodosi et al., 2018). Therefore, dust has been considered
95    as an excellent medium of long-range transport of organic matter (Falkovich et al., 2004).

There is a paucity of information's regarding the bioavailability of atmospheric organic carbon and his fate
in the ocean (Djaoudi et al., 2018). In this context, we investigate here the bioavailability to marine heterotrophic
bacteria of dissolved organic carbon (DOC) leached from aerosols. For this purpose, we performed a set of *in vitro*
biodegradation experiments in which a marine bacterial inoculum was exposed to water-soluble fractions of
100    anthropogenic and Saharan dust aerosols, as well as to glucose-amended and non-amended (control) treatments.

## 2.    Material and Methods

### 2.1. Aerosol sampling

Anthropogenic and Saharan dust aerosols were collected on Pall Flex® tissue-quartz pre-combusted (450°C,
6 h) filters (20.3 x 25.4 cm$^2$), by using high volume samplers (Tisch Environmental Inc., OH, USA). Anthropogenic
aerosols were collected during March 2016, in an urban zone in Marseille (South East of France: 43.3 °N and 5.4°E;
Fig. 1), during 15 days with an average flow rate of 1.42 m$^3$ min$^{-1}$ (Fig. 1). Saharan dust aerosols were collected during
June 2016 close to Medenine (South East of Tunisia: 33.3 °N and 10.5° E; Fig. 1), a major source of desert dust to the
Mediterranean Sea (Prospero et al., 2012), over periods of 24 h with an average flow rate of 0.7 m$^3$ min$^{-1}$ (Fig. 1).
High volume samplers were calibrated for flow rate just before sampling. Aerosol filters were individually wrapped
in a double pre-combusted aluminum foil and then stored at - 20 °C until the start of the experiment.

The content of total organic carbon (TOC) in collected aerosols was analyzed using a thermo-optical method
(EC/OC analyzer, Sunset Laboratories Inc.) on the basis of EUSAAR protocol (Cavalli et al., 2010). The amount of
DOC contained in the water-soluble fraction of both anthropogenic and Saharan dust aerosols was assessed by
leaching 1 cm$^2$ of filters in 30 mL of ultrapure water. After sonication during 40 min, it was filtered through a pre-
combusted (450 °C, 6 h) GF/F filter prior to analysis.

### 2.2. Experimental design

Bioassay experiments were run in triplicates (7 L each) on DOC-free artificial seawater in order to set aerosol-
derived DOC as the sole carbon source for marine heterotrophic bacteria. Artificial seawater was obtained by adding
pre-combusted NaCl (450 °C, 6 h) in ultrapure water to get a salinity of 36 g L$^{-1}$. The DOC concentration in the
artificial seawater was 6 µM. To avoid nutrient limitation, artificial seawater was enriched with nitrogen (NH$_4$Cl +
NaNO$_3$) and phosphate (KH$_2$PO$_4$), to final concentrations of 1 µM and 0.3 µM in the incubation bottles, respectively.

The experiment consisted of 3 treatments differing on the carbon source at a similar initial concentration.
Experimental treatments (i.e., N- and P-enriched artificial seawater) were amended with Saharan dust (D),
anthropogenic aerosols (A), and glucose (G) as a proxy of bioavailable carbon source. An unamended DOC treatment
(control; C) was run in parallel and consisted only on (N- and P-enriched) artificial seawater. As the amount of DOC
in the water-soluble fraction was lower in Saharan dust than anthropogenic aerosols  (Table 1), the amount of aerosol
added was set based on the C content in Saharan dust aerosols filter in order to fix a similar initial DOC concentration
among all amended incubation bottles (36 µM). To do so, particles from two whole Saharan dust filters and from 7.9





x 7.9 cm$^2$ of the anthropogenic aerosol filter were firstly leached, each in 650 mL ultrapure water. After being sonicated

for 40 min, the suspended particles were filtered through pre-combusted GF/F filters (450 °C, 6 h) to recover the dissolved fraction. A volume of 200 mL of each leachate was finally introduced in the corresponding aerosol amended treatments (D and A). The same DOC concentration (36 µM) was also set for the glucose- treatment.

A microbial inoculum was introduced in all incubation bottles. To prepare the microbial inoculum, surface seawater was collected at 5-m depth with a Niskin bottle at the MOOSE-Antares offshore station in the Mediterranean

Sea (42°48′ N, 6°10′ E; Fig. 1), on board R/V Téthys II. Seawater was first filtered on board through a 40-µm mesh plankton and then through 0.8-µm pre-cleaned (10% HCl and ultrapure water) polycarbonate filter to remove larger particles and plankton. Once in the laboratory, the microbial inoculum was further concentrated 30 times on a 0.2-µm pre-cleaned (10% HCl and ultrapure water) polycarbonate filter in a final volume of 180 mL. The DOC concentration in the 180-mL inoculum was 120 µM. A volume of 15 mL of the bacterial inoculum was added to each experimental

bottle. This approach minimizes the volume of seawater (and therefore the amount of DOC) added to the experimental bottles with the microbial inoculum (i.e. Lechtenfeld et al., 2015), allowing setting the aerosol-derived DOC as the main carbon source for heterotrophic bacteria. After dilution in the incubated volume (7 L), the contribution of marine DOC was < 0.3 µM. All experimental bottles were incubated in a controlled temperature room at 18 °C during 16 days in the dark and were regularly gently shaken.

**2.3. Subsampling and analyses**

Subsamples were collected from the experimental bottles at 13 selected times for heterotrophic bacterial abundance (BA) and heterotrophic bacterial production (BP) and at 3 selected times for ectoenzymatic activity (EEA), dissolved organic carbon (DOC), dissolved organic nitrogen (DON) and dissolved organic phosphorus (DOP) (see Table S1 for detail of subsampling).

Samples for BA (1.8 mL) were fixed with 18 µL of a preservative solution (Glutaraldehyde 0.25% final - Pluronic 0.01% final), kept during 15 min at room temperature in the dark and then transferred to a -80 °C freezer until analysis, within few days. Frozen samples were thawed at room temperature and were analysed using the FACSCalibur (BD Biosciences) flow cytometer (PRECYM flow cytometry platform, http://precym.mio.univ-amu.fr/). For BA cell counts, samples (0.3 mL) were incubated with SYBR Green II solution 1:10 (2 µL, Molecular Probes) for 15 min in

the dark, in order to stain the nucleic acids. Each cell was characterized by 3 optical parameters: light diffusion parameter side-scatter (SSC), green fluorescence (515-545 nm; SYBRgreen), and red fluorescence (670 LP; chlorophyll-a, in order to exclude autotrophic prokaryotes). Combining SYBRGreen fluorescence and SSC allowed to distinguish the cells from inorganic particles, detritus and free DNA (Marie et al., 2000). Data were processed using the CellQuest software (BD Biosciences), and BA was further determined using Summit 4.3 software (Beckman-

Coulter).

BP was estimated by $^3$H-leucine incorporation applying the centrifugation method (Smith and Azam, 1992). Samples (1.5 mL) were incubated in the dark between 1 and 6 h at 18 °C with a mixture of $^3$H-leucine (Perkin Elmer® specific activity 106 Ci mmol$^{-1}$) and non- radioactive leucine to a final concentration of 20 nM (6 nM $^3$H-leucine + 14 nM cold leucine) in 2-mL Eppendorf tubes. Incorporations were stopped by addition of trichloroacetic acid (TCA)





to a final concentration at 5%. The control was prepared for each treatment by the addition of TCA, before the addition of $^3$H-leucine. Samples were then centrifuged for 10 min at 16,000 g three times, first with the fixed sample, second with TCA 5% and finally with ethanol 80%. After resuspension of the pellet in 1.5 mL scintillation liquid (Ultima-Gold MV), radioactivity was determined by a liquid scintillation counter. Leucine incorporation rates were converted into carbon production using the conversion factors of 1.5 kg C per mole of leucine incorporated (Kirchman, 1993).

EEA were measured fluorometrically, using 2 fluorogenic model substrates, 4 methylumbelliferyl – βD-glucopyranoside (MUF- βglu) and L-leucine-7-amido-4-methyl-coumarin (Leu-MCA) as representative of β-glucosidase and aminopeptidase respectively (Hoppe, 1983). Hydrolysis rates were determined by incubating in black 24 microtiter-plate 2 mL of D and A treatments with 6 (0.05; 0.2; 1; 2; 5 and 10 µM) concentrations of each substrate model. To ensure linearity of the hydrolysis rate, the increase of fluorescence was measured at multiple time points

during the incubation period (ex/em: 380/440 nm for MCA and 365/450 nm for MUF, wavelength width 5 nm) in a VARIOSCAN LUX microplate reader. The instrument was calibrated with standards of MCA and MUF solutions diluted in < 0.2 µm filtered seawater.

Samples for DOC analysis (10 mL) were filtered online from the incubation bottles, through pre-combusted (450 °C, 6 h) 47-mm GF/F filters. The filtrates were then transferred into pre-combusted glass tubes where 50 µL of

phosphoric acid (H$_3$PO$_4$, 85%) was added. The glass tubes were sealed and stored in the dark at 4 °C until analysis. DOC was analyzed using the high temperature catalytic oxidation (HTCO) technique with a Shimadzu TOC-V analyzer. The daily calibration curve was based on 4 standard solutions (0, 50, 100 and 150 µM) of potassium acid phthalate (1,000 ng L$^{-1}$). The analytical precision of the procedure, based on 3-5 injections and analysis of the same sample, was 1.8% on average. The accuracy of the instrument was determined by the analysis of a deep seawater

reference (D. Hansell, Rosenstiel School of Marine and Atmospheric Science, Miami, USA). The average DOC concentration in the deep seawater reference was $45 \pm 3$ µM.

DON and DOP were determined after subtraction of the dissolved inorganic fraction of nitrogen (DIN) and phosphorus (DIP) from the total dissolved fractions (TDN and TDP), respectively. Samples for TDN, TDP, DIP and DIN (60 mL) were filtered online through pre-cleaned (HCl 10% and ultrapure water) 47-mm 0.2-µm polycarbonate

filters, then transferred into pre-cleaned (HCl 10% and ultrapure water) 60-mL HDPE bottles. Samples were kept frozen (-20 °C) until analysis through the conventional automated colorimetric procedure (Aminot and Kerouel, 2007). Prior to analysis, TDN and TDP were submitted to a wet oxidation according to Pujo-Pay and Raimbault (1994). The detection limits for DIN (NO$_3^-$ + NO$_2^-$) and DIP (PO$_4^{3-}$) analysis were 0.05 µM and 0.02 µM, respectively.

**2.4. Determination of labile DOC, bacterial growth efficiency and kinetic parameters**

The percentage of the labile dissolved organic carbon (LDOC) was calculated as:

$$LDOC(\%) = \frac{[DOC]_{initial} - [DOC]_{Final}}{[DOC]_{initial}} * 100$$



Bacterial growth efficiency (BGE, %) was calculated by dividing the time integrated BP by the corresponding labile fraction of DOC (LDOC).


$$BGE(\%) = \frac{Time\ integrated BP}{[\text{LDOC}]}\ x\ 100$$

Maximum hydrolysis rate ($V_{max}$) and Michaelis-Menten constant ($K_m$) were determined by fitting the EEA data using a nonlinear regression on the rectangular hyperbolic function following:

$$V = V_{max}\frac{S}{K_m + S}$$

Where V is the MUF- βglu or MCA-leu hydrolysis rate and S the concentration of the fluorogenic substrate.

**2.5. Statistics**

The three-replicate average values of bacterial activity (BA, BP and EEA) and nutrient concentrations (DOC, DON and DOP) in the G, D, A and C treatments were compared, over the incubation period, using a one-way ANOVA statistical test. ANOVA returns a p-value higher than 0.05 for the null hypothesis that the means of the different studied treatments are equal.

**3.   Results**

**3.1. Total and dissolved organic carbon content of aerosols**

The content of TOC in aerosols and of DOC in the water-soluble fractions were higher in anthropogenic (0.155 and 0.05 g C/g aerosols, respectively) than in Saharan dust aerosols (0.063 and 0.008 g C/g aerosol, respectively) (Table 1). Likewise, the dissolved fraction resulting from leaching was higher in the anthropogenic

aerosols (32%) than in Saharan dust (13%) (Table 1). One hour after aerosols amendments (T0), the measured initial concentration of DOC was 40 ± 3 µM in the G-treatment, 36 ± 2 µM in the A-treatment and 34 ± 3 µM in the D-treatment. Over the incubation period, DOC concentration decreased in all treatments (Fig. 2A). This decrease was highest in the G-treatment (22 ± 2 µM) followed by both D (9 ± 2 µM) and A (9 ± 4 µM) treatments and then the C-treatment (5 ± 1 µM) (Table 2; Fig. 2A). The resulted labile DOC fractions were 55 ± 2%, 25 ± 4% and 24 ± 8% in

G, D and A treatments, respectively (Table 2).

**3.2. Dissolved organic nitrogen and phosphate**

The initial DON and DOP concentrations in the C-treatment were 1.6 ± 0.1 µM and below the detection limit, respectively. The G-treatment exhibited the same initial DON concentration as the C-treatment, with values of 1.8 ± 0.2 µM while DOP concentrations reached 0.03 ± 0.01 (Fig. 1B, C). In A and D-treatments, the concentration of both

DON and DOP increased immediately after aerosol addition (T0). The DON concentration reached 4.3 ± 0.7 µM and 3.7 ± 0.2 µM in the A and D-treatments, respectively (Fig. 1A), resulting in lower C:N and higher N:P elemental ratios in A and D treatments than in G-treatment (Table S2). The increase in DOP after aerosol addition in the D-treatment was of 0.02 ± 0.005 µM and was similar to that observed in the G-treatment. In the A-treatment, the increase in DOP



was slightly higher reaching a value of $0.04 \pm 0.007$ (Fig. 2B), which resulted in lower C:P ratios in A than in both D
and G in the beginning of the experiment (Table S2).

In all treatments, over the incubation period, DON concentrations remained constant while DOP concentrations
exhibited a significant increase between T0 and T=5.7 days, to finally decrease to values below the detection limit at
the end of the incubation period (Fig. 1B and 1C). Over the incubation period, C:N elemental ratios decreased slightly
in A (ANOVA; $p<0.05$; F=5.3; df=8) and G (ANOVA; $p<0.05$; F=31.5; df=8) treatments while they remained constant
in the D (ANOVA; $p>0.05$; F=0.1; df=8) and C (ANOVA; $p>0.05$; F=0.5; df=8) treatments (Table S2). In contrast,
C:P and N:P elemental ratios were similar in the A-treatment while they decreased significantly in both D and G
treatments (Table. S2).

### 3.3. Microbial activity over the incubation period

In all the experimental bottles, an increase in both bacterial abundance (BA) and bacterial production (BP) was
observed during the incubation time. One replicate from the A-treatment diverged completely from the other replicates
(it did not show any bacterial growth and BP fluxes collapsed suddenly on day 6) and it was excluded from the data
processing.

At the beginning of the incubations (T0), BA was similar in all treatments (ANOVA; $p>0.05$; F=0.64; df=10),
with an average value of $2.3 \pm 0.1 \times 10^5$ cells mL$^{-1}$ (Fig. 3A). After 1.7 days of incubation, BA exhibited a 3-fold
decrease in all treatments. Following this decrease, a lag time phase was observed, during which BA continued to
decrease slowly in all treatments. This lag phase period was longer in the C-treatment (7.7 days), followed by both A
and D- treatments (5.7 days) and finally by the G-treatment (3.7 days; Fig. 3A). After that period, an exponential
growth of BA was observed first in the G-treatment, followed then by both A and D- treatments and finally by the C-
treatment (Table 3). During that exponential growth period, BA reached similar values (ANOVA; $p=0.34$; F=1.31;
df=10) in G ($6 \pm 3 \times 10^5$ cells mL$^{-1}$), A ($5 \pm 2 \times 10^5$), and D ($5 \pm 1 \times 10^5$) treatments (Fig. 3A). In the control treatment,
until t= 6.7 days, BA remained low, $< 0.6 \times 10^5$ cells mL$^{-1}$ and then increased up to $2.5 \pm 0.6 \times 10^5$ cells mL$^{-1}$ during
the exponential growth phase (Fig. 3A).  At the end of the incubation period, BA continued to increase in all treatments,
reaching up to twice the observed BA during exponential growth phase (Fig. 3A).

Initial BP was low in all experimental bottles, with an average value of $0.042 \pm 0.035$ ng C L$^{-1}$ h$^{-1}$ (Fig. 3B). BP
started to increase first in the G-treatment followed by D and then A treatments (Fig. 3B; Table 3). Like BA, BP also
exhibited an exponential growth phase, which started after a lag phase shorter than that of BA (Table 3). During the
exponential growth period (Table 3), BP growth rates were similar in G and D treatments (ANOVA, F=6.76; $p>0.05$;
df=5) and were significantly higher than in A and C treatments (ANOVA, F=10.47; $p<0.05$; df=10) (Table 3).
Likewise, maxima of BP reached were similar  in the D ($879 \pm 64$ ng C L$^{-1}$ h$^{-1}$) and G ($648 \pm 156$ ng C L$^{-1}$ h$^{-1}$)
treatments (ANOVA, F=5.63; $p=>0.05$; df=5) and were significantly higher (ANOVA, F=43.4; $p<0.05$; df=10) than
in A ($124 \pm 39$ ng C L$^{-1}$ h$^{-1}$) and C ($120 \pm 56$ ng C L$^{-1}$ h$^{-1}$) treatments. In the C treatment, until t=6.7 days, BP was low
($< 2$ ng C L$^{-1}$ h$^{-1}$) and then increased at the end of the incubation period (Fig. 3B).



The Michaelis Menten fit of aminopeptidase and β glucosidase activities were significant only at the end of the incubation period (T=15.7 days). Thereby, only these data are presented (Fig. S1). The average aminopeptidase $V_m$ in

the D-treatment was 1.3 ±0.2 nM h$^{-1}$ and was significantly lower by approximately 4 times (ANOVA, F=34.3; p<0.05; df=4) than that observed in the A-treatment (4.4 ± 1.0 nmol h$^{-1}$). β-glucosidase $V_m$ was much lower than that of aminopeptidase in both A and D treatments. At the opposite to what was obtained with MCA-leucine, and although not significantly different due to the variability within triplicates (ANOVA, F=6.24, p>0.05, df=4), the average β-glucosidase $V_m$ was slightly higher in D, with values ranging 0.08-0.16 nM h$^{-1}$ compared to 0.02-0.03 nM h$^{-1}$ in A.

These differences between treatments were still observed considering specific $V_m$ (per bacterial cell, Fig. 4). Indeed, like for volumetric $V_m$ rates, cell specific aminopeptidase $V_m$ (Fig.4) were lower in D than A (ANOVA, F=650.5; p<0.05; df=4). Likewise, cell specific β glucosidase $V_m$ showed slightly higher values in D (0.06 - 0. 16 x 10$^{-18}$ mol cell L$^{-1}$ h$^{-1}$) compared to A (range 0.015 - 0.026 x 10$^{-18}$ mol cell L$^{-1}$ h$^{-1}$), although not significantly different between the two aerosols treatments (Fig. 4). The opposite trend was noticed for β glucosidase $K_m$, with lower values in D

(0.06 - 0.16) than A (0.4 - 2.3) treatment.

The bacterial growth efficiency (BGE) in the D-treatment (14.2 ± 5.5%) compared well with the G-treatment (7.6 ± 2%) (ANOVA, F=3.9; p>0.05; df=5) and both were significantly higher than that observed for the A-treatment (1.7 ± 0.1%) (ANOVA, F=7.91; p=0<0.05; df=5) (Table 3).

## 4. Discussion

### 4.1. Influence of the aerosol source on marine heterotrophic bacterial activity

In this study, the addition of Saharan dust and anthropogenic aerosols, as a sole C-source to marine bacteria stimulated both heterotrophic bacterial abundance (BA) and production (BP), evidencing the bioavailability of aerosol-derived DOC.

However, at the beginning of the experiment, a lag time period was observed in all treatments, during which

BA dropped by 3 times of its initial value. This consistent decrease of BA may be attributed to a stress of marine heterotrophic bacteria during the preparation of the inoculum and to their contact with new environment/matrix, containing an unusual source of carbon. Indeed, although a lag time period was also recorded in the G- treatment, it was shorter than that in A and D-treatments, which could suggest a non-immediate bioavailability of aerosol organic matter compared to glucose. This non-immediate bioavailability of the dissolved pool of organic matter has been

highlighted in the ocean in a review on marine DOM incubation experiments (Lønborg and Alavrez-Salgado, 2012; Sipler and Bronk, 2015 and references therein). Other factors including potential additions of toxic molecules and/or viruses from the aerosol leachate could have exerted a control on marine heterotrophic bacteria at the beginning of the experiment, explaining the observed lag time period.

After the lag phase, an increase in BA and BP was observed first in the G-treatment, followed by D and then A-

treatment, with bacterial growth rates significantly higher in G and D- treatments than in A-treatment. In contrast to Marín et al. (2017b), in which BA and BP exhibited a contrasted response following mineral dust and anthropogenic aerosols addition, the observed increase of both BA and BP in this study were higher in the D-treatment than in A. Following this increase in bacterial activity, DOC was consumed, highlighting a higher LDOC fraction in the G-



treatment than in A and D-treatments. In those two treatments, LDOC fractions were quantitatively similar, despite a
higher solubility of organic carbon derived from anthropogenic aerosols compared to Saharan dust aerosols.
Nevertheless, contrasted metabolic pathways were evolved by marine heterotrophic bacteria in A and D treatments
over the time scale of the incubation period. Indeed, the BGE in the D- treatment ($14.2 \pm 5.5\%$) compares well with
that of the G treatment ($7.6 \pm 2\%$) and both values were significantly higher than that of the A- treatment ($1.7 \pm 0.1\%$).
This result suggests that, in the experimental conditions of this study, the metabolic use of LDOC in G and D-
treatments was energetically equivalent, with evidences of approximatively 1/10 of DOC incorporated going into
structural components. In contrast, the carbon consumed in A-treatment was mostly catabolized.

The factors controlling whether organic carbon is catabolized or incorporated into microbial biomass is still poorly
resolved. Nevertheless, the conversion of carbon into biomass only occurs after non-growth requirements have been
satisfied and sufficient excess of carbon and energy are available (Del Giorgio and Cole, 1998). To explain the
differences in BGE observed in this study among glucose, Saharan dust and anthropogenic aerosol amendments,
several hypotheses can be proposed (see below).

### 4.2. Potential controlling factors of the bacterial growth efficiency

A number of parameters were controlled at the beginning and/or during the incubation period and can thus be
reasonably excluded. Indeed, temperature and initial microbial inoculum were equal among all treatments. Since DIN
and DIP were added in excess, N and P availability were not growth limiting over the incubation period. In contrast,
DOM composition or quality and/or micronutrients availability (i.e. Fe, Zn) may have influenced the BGE in A and
D aerosols treatments, as they are reported as the main variables controlling the BGE in fresh and marine waters
(Fenchel and Blackburn, 1979; Biddanda et al., 1994; Carlson and Ducklow, 1996; Kirchman et al., 2018 and
references therein).

In this study, BGE was lower in A than in both D and G treatments, which could suggest a DOM pool of a lower
quality in anthropogenic aerosols with respect to Saharan dust. The observed differences in ectoenzymatic activities
between D- and A-treatments support this hypothesis. The higher cell specific aminopeptidase activity in A treatment
combined with its enzymatic system adapted to low concentrations (low $K_m$), presumably suggest a lower bacterial
access to amino acids despite a higher DON concentration in A than in D treatment. Thus, the versatility in catabolic
enzymatic synthesis was probably limited due to less access to amino acids, contributing to the observed low BGE in
the A treatment. In the D treatment, the lower development of aminopeptidase together with a better development of
β-glucosidase capacities (higher specific $V_m$; lower $K_m$ and thus a better affinity for substrate) suggested that molecules
derived from carbohydrates were used for anabolism and as a source of energy in one hand and that organic nitrogen
source were more available to heterotrophic bacteria in the other hand, leading to a better global benefit for
heterotrophic bacteria as shown by the higher BGE.

The observed differences in ectoenzymatic activities may have been potentially upregulated by trace metals
and/or vitamins cofactors on one hand, or inhibited due to a supply of toxic elements, in the other hand. Saharan dust
deposition has been suggested as an important source of iron to surface waters (Jickells et al., 2005). There is an
increasing evidence on the potential role of iron as a cofactor requirement for several enzymes involved in
photosynthesis, $N_2$ fixation and remineralization (Wu et al., 2000; Twining et al., 2004; Browning et al., 2017). In this



study, a potential supply of iron or other trace metal elements (i.e. zinc, cobalt) following Saharan dust aerosols addition may have stimulated the bacterial ectoenzymatic activity and/or other metabolic pathways in the D-treatment. At the opposite, an eventual upload of toxic elements, following anthropogenic aerosols (Paytan et al., 2009; Jordi et al., 2012), may have constrained the bacterial biosynthesis in the A-treatment.

Besides micronutrients, a significant amount of dissolved organic nitrogen (DON) was leached from both A and D aerosols, highlighting the important contribution of not only C, but also N containing organic molecules to atmospheric deposition, such as previously reported through atmospheric flux quantification and modeling (Markaki et al., 2010; Djaoudi et al., 2018; Kanakidou et al., 2012; 2018; Galleti et al., 2020, this special issue). This DON supply resulted in a significant decrease of C:N elemental ratios in both A and D treatments with respect to G,
immediately after seeding (Table S2). BGE of natural assemblages of marine bacteria grown on a range of substrates has been shown to be inversely related to the C:N ratio of the substrate. Indeed, Goldman al. (1987) showed that the BGE was independent of the source of C and N but increased as the C:N ratio of the substrate decreased. In this study, although C:N stoichiometric ratios were similar in D and A aerosols treatments over the incubation period, contrasted BGE were observed suggesting that elemental ratios alone were not sufficient to explain the differences between A
and D treatments, in the condition of the experiment.

The contrasted BGE and ectoenzymatic development toward access to organic molecules in both D and A treatments may have been linked to the primary sources and the chemical composition and/or structure of aerosols as well. Anthropogenic aerosols are associated with various combustion processes, including industrial production; vehicles exhaust and domestically or waste burning, producing mainly soot particles (Li et al., 2001; Alves et al.,
2012; Kanakidou et al., 2012; 2018). These soot particles are generally coated with various organic compounds, forming complex mixtures of highly condensed organic matter. For example, polycyclic aromatic hydrocarbons and a large number of aliphatic compounds, are ubiquitous within anthropogenic aerosols (Omar et al., 2006; Alves et al., 2012; Barhoumi et al., 2018). However, these compounds have low water-solubility, and thus their contribution to the DOC pool could not explain alone the low BGE observed in the A-treatment with respect to both D and G-treatments.

In Mediterranean big cities, the atmospheric organic fraction is also related to secondary organic aerosols (SOA) formation, with a contribution to the fine fraction of aerosols, ranging between 60 and 80% (Amato et al., 2015). In urban atmospheric aerosols, the water-soluble organic carbon has been highlighted as mostly aliphatic in nature (approximately 95% by C mass), with major contributions from alkyls and oxygenated alkyls (~ 80%), carboxylic acids (~ 10%), and to a lower extent from aromatic functional group (~ 4%). Among the organic species
found in anthropogenic aerosols, alkyls or oxygenated alkyls, (poly)carboxylic acids/carboxylates (e.g. formate and oxalate), sugars (e.g. levoglucosan) and HUmic-LIke Substances (HULIS) have been reported as the main contributors (Jaffrezo et al., 2005; Sannigrahi et al., 2006; Salma et al., 2013; Theodosi et al., 2018). Recently, HULIS, which are a mixture of high molecular weight organic (hydrophobic aliphatic and aromatic) compounds has been highlighted as contributing for a significant proportion of water-soluble organic carbon (up to 70%) (Zheng et al., 2013; Violotis et
al., 2017).Thus, the HULIS compounds potential contribution to the DOC pool in the A-treatment, may have constrained the overall bioavailability of energy and carbon, and thus the potential of marine bacteria for biosynthesis.



Otherwise, an enrichment of anthropogenic aerosols by organic compounds such as formate, acetate, oxalic and malonic acids has been previously reported in the atmosphere close to emission sources or after atmospheric transport (Sullivan and Prather, 2007; Leaitch et al., 2009; Paris et al., 2010). Some of these compounds (i.e. formate, malonic acids) as being involved in microbial metabolic pathways, could promote respiration and thus a low BGE.

The most abundant organic compounds in dust aerosols revealed a vegetal origin and are more particularly highlighted as the reflect of the past vegetation cover of soils, from where dust aerosols were emitted (Eglinton, 2002). Thus, organic carbon was associated with more or less decomposed biological residues, including micro-organisms debris, as well as humic substances (Conen et al., 2011). In this study, Saharan dust aerosols were locally sampled within the short time scale of the dust event (48 h). This may prevented from an internal mixing of Saharan dust aerosols with other organic compounds in the atmosphere, revealing a higher bioavailable character of the Saharan dust derived-DOC for marine bacteria, when compared to the anthropogenic aerosols derived-DOC. Thereby, the higher BGE in D treatment with respect to A may have been the result of a DOC pool of a vegetal origin, more available to marine bacteria than that derived from anthropogenic emissions. For instance, a probable richness in cellulose – containing molecules in the "fresh" dust used in this study could explain the higher β glucosidase activity in the D-treatment.

### 4.3. Potential implications for the marine carbon cycle in the Mediterranean Sea

Atmospheric fluxes of DOC in the Mediterranean Sea have been reported to be 6 times higher than river fluxes (Djaoudi et al., 2018; Galleti et al., 2020 this issue), highlighting atmospheric deposition as a major allochthonous source of carbon to this marine region. Hereafter, based on these previously reported atmospheric DOC fluxes in the Mediterranean Sea (Djaoudi et al., 2018; Galleti et al., 2020) and on the partitioning between labile (LDOC) and refractory DOC (RDOC) in aerosols obtained in this study, we propose an estimation of LDOC and RDOC fluxes from both Saharan dust and anthropogenic deposition and their potential contribution to marine carbon cycle.

Although atmospheric deposition is a mixture of different particles from several sources (i.e. anthropogenic, Saharan dust and/or biomass burning), to give a rough estimation of RDOC and LDOC fluxes we will assume that contribution of LDOC and RDOC obtained in this study are valid for previous quantifications of atmospheric DOC fluxes (59-120 mmol m$^{-2}$ year$^{-1}$) (Djaoudi et al., 2018; Galleti et al., 2020). Based on LDOC fractions obtained in this study, atmospheric fluxes of LDOC ranging between 13-36 mmol C m$^{-2}$ year$^{-1}$ and between 11-32 mmol C m$^{-2}$ year$^{-1}$ would be brought from Saharan dust (LDOC: 22-30%) and anthropogenic deposition (LDOC: 19-27%), respectively. Beside the labile fraction, at the time scale of the biodegradation experiment conducted in this study, a large fraction of the initial DOC concentration derived from both Saharan dust (70-78%) and anthropogenic aerosols (73-81%) remained non-consumed, suggesting a significant input of semi-labile and/or refractory DOC into seawater from aerosols dissolution.

In the Mediterranean Sea, bacterial carbon demand cannot be satisfied solely nor with DOC of autochthonous origin (*i.e.* for instance phytoplankton DOC extracellular release, Anderson and Turley; 2003), neither with terrestrial riverine DOC fluxes (Van Wambeke et al., 1996; Del Giorgio and Cole, 1997), emphasizing a critical lack in our understanding of sources and sinks of carbon, in marine systems. Indeed, using a BGE of 10%, Van Wambeke et al. (2020, this special issue) examined autochthonous sources of C through macromolecule hydrolysis and showed that



the cumulated hydrolysis rates of C-proteins and C-polysaccharides contributed for a small amount to the bacterial
carbon demand, on average $2.5 \pm 1.3\%$ in the epipelagic waters during the spring period. Similarly, measured rates of
photosynthetic DOC production represented only 25% of estimated bacterial carbon demand in the Mediterranean Sea
(Marañón et al., 2020; this special issue). Thus, in such oligotrophic areas, atmospheric deposition of DOC might be
one of the missing pieces constraining our understanding of the biogeochemical cycling of carbon, and its labile
fraction would be one possible means of contribution to the bacterial carbon demand. To get insight on this potential
contribution, atmospheric fluxes of LDOC could be compared to photosynthetic DOC production rates. Such rates are
obtained generally during 24h-long incubations with 14C bicarbonate. It includes not only direct phytoplankton
extracellular release, but also excretion from grazers and byproducts from viral lysis during the incubations. In the
Mediterranean Sea, such fluxes ranged between 4.2 and 10.8 mmol C $m^{-2}$ $day^{-1}$ (López-Sandoval et al., 2010). By
assuming the percentages of LDOC observed in our biodegradation experiment as representative for all Saharan dust
and anthropogenic aerosols, and are valid for previously reported atmospheric fluxes (0.03-1.78 mmol C $m^{-2}$ $day^{-1}$)
(Djaoudi et al., 2018; Galleti et al., 2020 this special issue), daily atmospheric fluxes of LDOC would range between
7-534 mmol C $m^{-2}$ $day^{-1}$ for Saharan dust deposition and between 6-480 mmol C $m^{-2}$ $day^{-1}$ for anthropogenic
deposition. Assuming photosynthetic DOC is fully labile and one of the major autochthonous LDOC source, Saharan
dust deposition fluxes would increase marine LDOC production rates of 2-13%, and similarly anthropogenic
deposition would result in an increase of 1.5-12%, supporting the hypothesis of a potential role of atmospheric
deposition of DOC in sustaining secondary production.

The contrasted metabolic pathways evolved by marine heterotrophic bacteria whether they are facing
anthropogenic or Saharan dust aerosols, would potentially influence the fate of derived DOC. When facing
anthropogenic aerosols, the result of the present study indicated that most of the consumed carbon is catabolized (BGE
$= 1.7 \pm 0.1\%$), and thus would be returned to the atmosphere as $CO_2$. However, when heterotrophic bacteria are facing
Saharan dust aerosols, approximatively 1/10 of the LDOC was incorporated into biomass, resulting in a BGE of 14.2
$\pm 5.5\%$. Nevertheless, atmospheric inputs of labile organic matter can lead to increased remineralization of the marine
DOM, potentially acting as a priming effect. Likewise, the marine dissolved organic matter pool, by shaping
bacterioplankton composition, could likely influence the microbial utilization of anthropogenic and Saharan dust
derived organic matter as well, thus influencing BGE. Therefore, investigating the interplay between the different
sources of DOM in link with microbial activity would allow getting further insight regarding the striking interaction
between atmospheric deposition and the marine carbon cycle, particularly regarding the role of marine bacteria as a
link or sink of carbon.

Aside a potential contribution of atmospheric DOC deposition in sustaining secondary production, the semi-labile
and/or refractory fractions of those atmospheric inputs, would influence the ocean surface biogeochemistry as well as
carbon export. DOC export has been highlighted as playing a critical role within the biological carbon pump in the
Mediterranean Sea, leading to an export of 17 g C $m^{-2}$ $year^{-1}$ (Guyennon et al., 2015). Estimated as for LDOC, RDOC
fluxes would range between 0.5-1.1 g C $m^{-2}$ $year^{-1}$ for Saharan dust deposition and between 0.5-1.2 g C $m^{-2}$ $year^{-1}$ for
anthropogenic deposition. Thereby, the contribution of refractory DOC derived from Saharan dust and anthropogenic



deposition would contribute to the DOC export of 3-6.5% and 3-7%, respectively, likely influencing the biological carbon pump especially in increasing scenarios of atmospheric emissions.

## 5.  Conclusion

Organic carbon derived from anthropogenic aerosols exhibited a higher solubility (32%) with respect to
Saharan dust (13%). Despite such a difference, the amount of bioavailable dissolved organic carbon (DOC) to marine heterotrophic bacteria was quantitatively similar, with contributions of the labile dissolved organic carbon to the total dissolved organic fraction of $25 \pm 4\%$ and $24 \pm 8\%$ in Saharan dust and anthropogenic aerosols, respectively. Interestingly, the bacterial growth efficiency (BGE) in the Saharan dust treatment ($14.2 \pm 5.5\%$) was higher than that of the anthropogenic treatment ($1.7 \pm 0.1\%$), suggesting differences in the metabolic response depending on the aerosol
source. This study reveals a new link between atmospheric deposition and the oceanic carbon cycle. Indeed, inputs of atmospheric anthropogenic carbon to the ocean, could promote its respiration by bacterial communities. In contrast, carbon derived from Saharan dust aerosols may contribute to biomass production.  These results question the future trajectory of ocean-climate feedbacks in oligotrophic oceans, particularly in increasing scenarios of anthropogenic emissions.

**Acknowledgment**

This work is a contribution to the Labex OT-Med (n° ANR-11-LABX- 0061) funded by the French Government «Investissements d'Avenir» program of the French National Research Agency (ANR) through the A∗MIDEX project (n° ANR-11-IDEX-0001-02). This study was conducted as part of the WP4 Mermex/ Mistral project and is a contribution to the PEACETIME research project and to the international SOLAS program. This work received the
support from the Institut des Régions Arides (IRA) from Médenine, Tunisia, as well as the IRD French-Tunisian International Joint Laboratory (LMI) "COSYS-Med". The authors also acknowledge Jean-Louis Rajot (IRD, iEES, Paris) for his constructive comments. The authors would like to thank PRECYM Cytometry platform. The associated Editor in Chief is acknowledged for her relevant comments and suggestions on the manuscript.







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





**Table 1. Total organic carbon (TOC) and dissolved organic carbon (DOC) content in anthropogenic and Saharan dust aerosols used for bacterial incubations.**

| Aerosols | TOC [g C/ g aerosols] | DOC [g C/ g aerosols] | Water solubility [%] |
|---|---|---|---|
| Anthropogenic | 0.155 | 0.05 | 32 |
| Dust | 0.063 | 0.008 | 13 |

**Table 2. Dissolved organic carbon decrease and its contribution to the initial DOC stocks (expressed as a percentage of labile dissolved organic carbon, LDOC).**

| | DOC decrease; µmol C L-1 | LDOC [%] |
|---|---|---|
| C1 | 5 | 26 |
| C2 | 6 | 26 |
| C3 | 4 | 25 |
| **Mean (± SD)** | **5 ± 1** | **26 ± 1** |
| G1 | 19 | 53 |
| G2 | 24 | 57 |
| G3 | 22.5 | 54 |
| **Mean (± SD)** | **22 ± 3** | **55 ± 2** |
| A1 | 6 | 27 |
| A2 | 11 | 19 |
| **Mean (± SD)** | **9 ± 4** | **24 ± 8** |
| D1 | 7 | 22 |
| D2 | 8.5 | 24 |
| D3 | 11 | 30 |
| **Mean (± SD)** | **9 ± 2** | **25 ± 4** |



**Table 3. Duration of the exponential growth phase for BA and BP, bacterial growth rate (μ) estimated from BP changes during the exponential growth phase, time integrated bacterial production (BP) during the exponential growth phase and bacterial growth efficiency (BGE). Data from each triplicate and average (± SD) values are given for the control (C), glucose, anthropogenic and Saharan dust treatments.**


| Treatments | Exponential phase period for BA [days] | Exponential phase period for BP [days] | $\mu$; [d$^{-1}$] | Time integrated BP[b] [μmol C L$^{-1}$ dt] | BGE $_{BP}$ [%] |
|---|---|---|---|---|---|
| C1 | 7.7 – 13.7 | 1.7 - 8.7 | 1.23 ± 0.19 | 0.23 | 5 |
| C2 | 6.7 - 8.7 | 1.7 - 8.7 | 0.94 ± 0.15 | 0.11 | 2 |
| C3 | 6.7 – 8.7 | 1.7 - 8.7 | 1 ± 0.13 | 0.18 | 4.5 |
| **Mean (± SD)** | | | **1.05 ±0.15** | **0.17 ± 0.06** | **4 ± 2** |
| G1 | 3.7 - 5.7 | 0 – 5.7 | 1.99 ± 0.32 | 1.07 | 5.6 |
| G2 | 2.7 – 4.7 | 0.7 – 3.7 | 3.44 ± 0.52 | 1.87 | 7.8 |
| G3 | 2.7 – 3.7 | 0 – 3.7 | 3 ± 0.41 | 2.16 | 9.6 |
| **Mean (± SD)** | | | **2.81 ± 0.74** | **1.70 ± 0.57** | **7.6 ± 2** |
| A1 | 5.7 – 8.7 | 2.7 – 6.7 | 1.64 ± 0.21 | 0.11 | 1.8 |
| A2 | 4.7 – 8.7 | 2.7 – 6.7 | 1.62 ± 0.28 | 0.19 | 1.7 |
| **Mean (± SD)** | | | **1.63 ± 0.01** | **0.15 ± 0.05** | **1.7 ± 0.1** |
| D1 | 5.7 – 7.7 | 2.7 – 7.7 | 1.74 ± 0.09 | 1.11 | 16.1 |
| D2 | 5.7 – 7.7 | 0.7 – 7.7 | 1.41 ± 0.10 | 1.58 | 18.5 |
| D3 | 4.7 – 5.7 | 1.7 - 5.7 | 1.80 ± 0.30 | 0.88 | 8.1 |
| **Mean (± SD)** | | | **1.65± 0.20** | **1.19 ± 0.36** | **14.2 ± 5.5** |





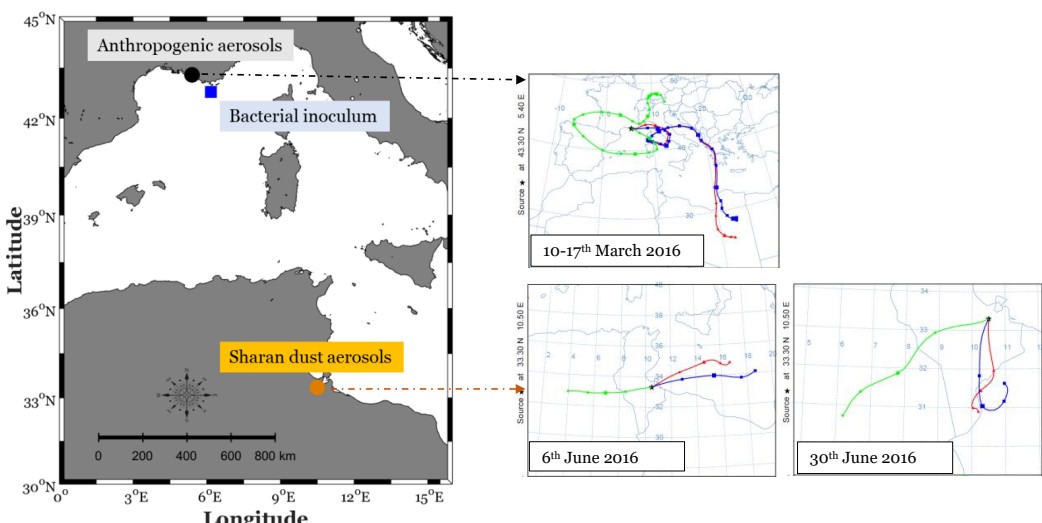

**Figure 1. Sampling stations for anthropogenic aerosols (Marseille, South East of France) and Saharan dust aerosols (Medenine, South East of Tunisia), as well as for bacterial inoculum (MOOSE-Antares offshore station in the northwestern Mediterranean Sea). Backward air mass trajectories are represented for the periods during which anthropogenic (10-17 March 2016) and Saharan dust aerosols (6th and 30th June 2016), using the HYSPLIT trajectory model (https://www.ready.noaa.gov/hypub-bin/trajtype.pl?runtype=archive).**



**Panel A**

**Panel B**


**Panel C**

**Figure 2. Changes in dissolved organic carbon (DOC, panel A) nitrogen (DON, panel B) and phosphate (DOP, panel C) concentrations over the incubation period in glucose (left, green bars), anthropogenic aerosols (middle, gray bars) and Saharan dust aerosols (right, orange bars) treatments. The control is plotted in each panel (blue bars).**





**Panel A**

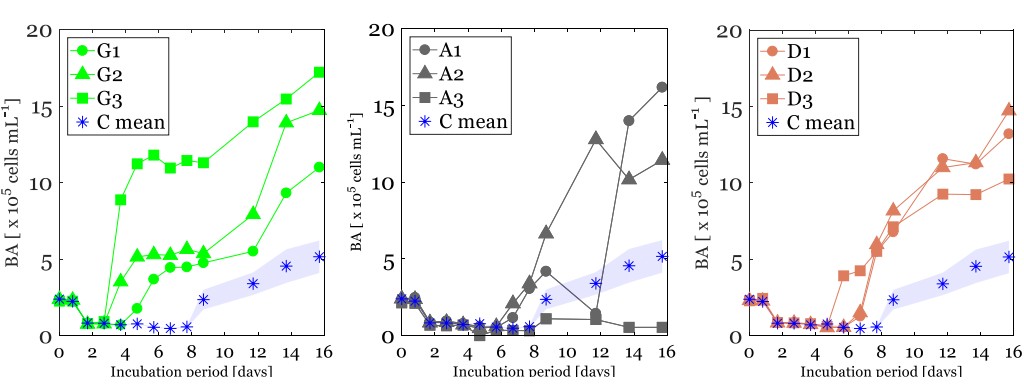


**Panel B**

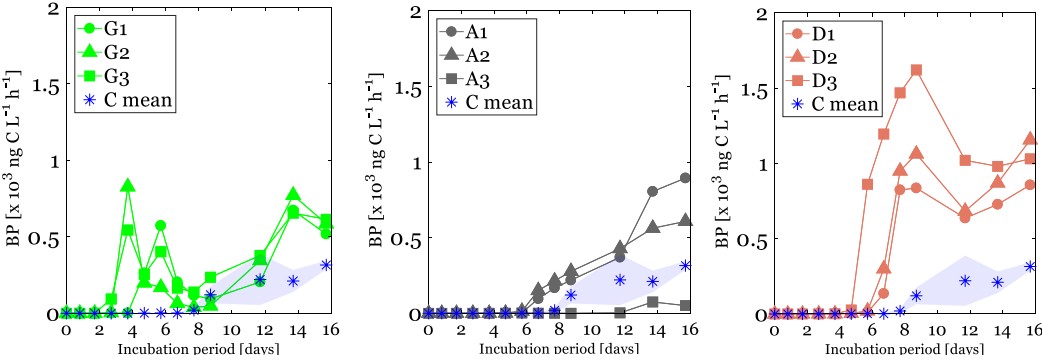

**Figure 3. Changes in heterotrophic bacterial abundance (BA; panel A) and production (BP, panel B) over the incubation period in glucose (left, green plots), anthropogenic aerosols (middle, gray plots) and Saharan dust aerosols (right, orange plots) treatments. The control is plotted in each panel: the average value is represented by the stars and the standard deviation by the bounded lines.**







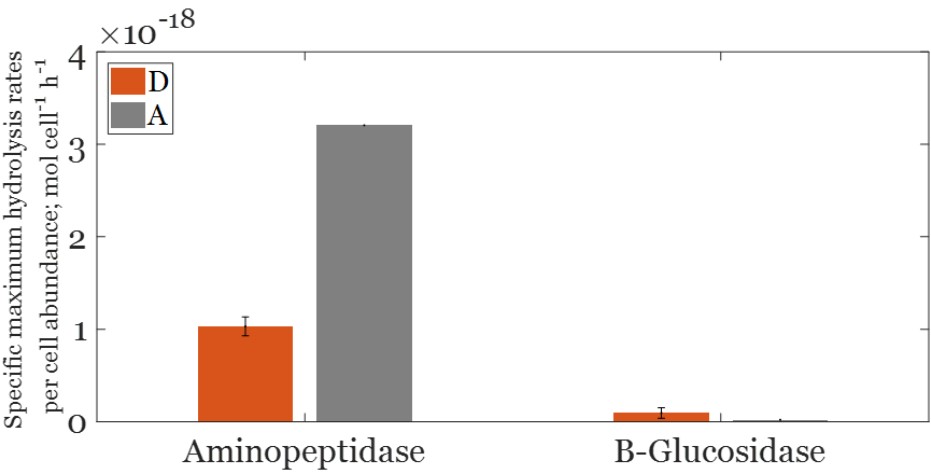


**Figure 4. Cell specific maximum aminopeptidase and β glucosidase hydrolysis rates per cell abundance in Saharan dust (orange bars) and anthropogenic (gray bars) aerosols treatments, at the end of the incubation period.**



