# Peer review of "Potential bioavailability of organic matter from atmospheric particles to marine heterotrophic bacteria"

_Biogeosciences, 2020_

## Referee Comment (RC1) · Mariana Bernardi Bif (Referee) · 17 Aug 2020

The original manuscript entitled "Experimental evidence of the potential bioavailability for marine heterotrophic bacteria of aerosols organic matter" investigates the bioavailability of organic carbon from aerosols to marine heterotrophic bacteria in the Mediterranean Sea. The authors performed dark incubations using natural microbial assemblages exposed to three different treatments under similar initial organic carbon concentrations (glucose solution, Saharan dust-derived carbon and Anthropogenic-derived carbon) and one control treatment (artificial seawater amended with inorganic nutrients). The organic carbon contained in the dust was dissolved into dissolved organic

carbon prior to the experiments. Although the bioavailability of organic carbon from the Saharan dust treatment was similar to the Anthropogenic treatment, the quality of organic carbon was different and reflected on the microbial metabolic response. The bacterial growth efficiency was higher in the Saharan dust treatment than the Anthropogenic treatment, suggesting that organic matter derived from Saharan dust had a higher quality with a significant portion being incorporated into biomass. On the other hand, most of the carbon consumed in the Anthropogenic treatment was catabolized. Interestingly, this contrasting outcome has local impact in the marine carbon cycle, since more or less contribution from the two sources will either stimulate biomass or respiration, ultimately contributing to the carbon pump efficiency.

Overall, the manuscript is very well organized and written. The introduction is complete with a fair literature revision that points out the lack of studies involving the lability of organic carbon from aerosols to marine microbes. The methodology is well developed and it did not raise any concerns from my side. The outcome is really interesting and will for sure contribute to the marine biogeochemistry field. I do have a few comments that would overall contribute to the manuscript and will be addressed below.

1 – The manuscript title is a bit too long, consider rewriting. Two suggestions: "Experimental evidence of the potential availability of organic matter from aerosols to heterotrophic bacteria" or "Potential bioavailability of organic matter from aerosols to heterotrophic bacteria".

2- I would be careful when addressing DOC budgets, extensively mentioned in the introduction and discussion, and comparing with budgets estimated from the experiments. The dust sources used in this experiment were artificially dissolved prior to incubation, but they arrive in the environment as POC. Is it realistic that, in the environment, dust particles would stay enough time at the ocean's surface to be dissolved to that proportion found in Table 1 before sinking? This could be addressed better in the discussion.

3- Introduction: in Line 65: "Nutrient availability and microbial community structure regulate the accumulation and the remineralization of DOM, influencing export efficiency". I do not feel comfortable suggesting papers of mine, but we addressed this issue through incubation experiments with nutrient amendments and observed DOC and DON dynamics. Bibliography: Bif, M.B.; Hansell, D.A.; Popendorf, K.J. Controls on the fate of dissolved organic carbon under contrasting upwelling conditions. Frontiers in Marine Science, v.5 (463), 2018. I think this can contribute to both introduction and discussion sections.

In Lines 76-88: Although atmospheric dust is of undoubted importance in oligotrophic regions of the Atlantic and Mediterranean Sea, it does not play a role in other oligotrophic regions such as in the South Pacific Subtropical Gyre. This could be better addressed by adding one or two sentences as it gave the impression that dust deposition is important in every oligotrophic region. Example of bibliography: Jickells, T. D., et al. 2005. Global iron connections between desert dust, ocean biogeochemistry and climate, Science, 308, pp.67–71.

4- 2.2.Experimental design:

In the control and glucose treatment, why didn't the authors add Fe to the solution? This is a limiting nutrient for heterotrophic bacteria, is probably found in very high concentrations in the dust treatments and could make a difference in the C and G incubations.

In the paragraph starting in l.122, was DOC analyzed using the method described in the paragraph starting in l.178? Please clarify.

Line 178: Replace "online from" with "in line with".

5- 3.Results:

In section 3.1. I see an overall DOC increase in the C treatment instead of a $5\mu M$ decrease, observed in Figure 2, Panel A.

---

## Referee Comment (RC2) · Anonymous Referee #2 · 24 Aug 2020

This manuscript reports and contextualizes the impact of atmospheric nutrient additions (water soluble fraction, WSF) of natural (Sahara dust, D) and anthropogenic (A) origin on the heterotrophic bacteria of the surface Mediterranean Sea. The experimental approach consisted on dark microbial incubations in artificial seawater with controlled additions of atmospheric nutrients and a common bacterial inoculum. Apart for the two atmospheric treatments (D and A), glucose (G) and control (C) treatments were also conducted for comparison. Chemical (DOC, DON, DOP) and microbial parameters (BA, BP, EEA) were followed during 2 weeks.

The experimental approach is adequate and the results –contrasting LDOC (%) and

BGE (%)– are interesting and potentially relevant in the context of the increasing fluxes of organic nutrients from the atmosphere to the surface ocean, particularly in oligotrophic seas close to densely populated areas (e.g. the Mediterranean Sea). However, there are some issues that should to be clarified / commented / discussed by the authors:

1) Inorganic nutrients were added to the microbial cultures to avoid inorganic nutrient limitation. Final concentrations were 1 uM-N and 0.3 uM-P, which are well above the expected concentration in the surface layer of an oligotrophic region. Given that inorganic nutrients have been added in the same concentrations to all treatments, comparison among treatments is valid but… what about extrapolation of your results to oligotrophic conditions in the field?;

2) The WSF of Sahara dust and anthropogenic aerosols do not contain only DON and DOP but also inorganic N (ammonium, nitrite and nitrate) and phosphate. Therefore, even if you have not added inorganic nutrients to the microbial cultures (see point 1), inorganic N and P would be added in the WSF. In this regard, it is relevant that the concentrations of inorganic N and P in the WSF are presented and discussed;

3) It is reported that the DOC concentration in artificial seawater was 6 uM and in the WSF <0.3 uM (once diluted in the ASW). However, the average DOC concentration in the control treatments was 19 uM (calculated from Table 2). What caused this difference? Should we assume that it also occurs in treatments G, A and D?;

4) The estimates of LDOC(%) are obtained for the 4 treatments (C, G, A and D) comparing the initial DOC and the DOC decrease of each treatment. However, a considerable part of the initial DOC is already present in the control treatment (see point 3). Therefore, how should we interpret the LDOC(%) numbers in Table 2? For example, if the DOC decrease in the control treatment is 5 uM and in the G treatment is 22 uM, the DOC decrease exclusively due to the glucose addition should be 17 uM. Given that the DOC of the glucose addition is 40 uM and in the control is 19 uM, the LDOC(%)

[Figure]

of the glucose addition would be 81% (=(22 − 5) / (40 − 19)). This is very different from the 55% in Table 2. For treatments A and D the difference is not so large but it is conceptually important;

5) The same reasoning is applicable to the BGE(%) calculations (LDOC is in the denominator of the formulae). For example, for the G treatment BGE(%) should be 9% (=(1.7 − 0.7)/ (22 − 5) and for the D treatment 26% (=(1.19 − 0.17)/(9 − 5)). It really makes a difference;

6) Also concerning BGE (%), it should be better to use the bacterial biomass, calculated from BA with a conversion factor, rather than BP; and

7) Extrapolation of your results to the entire Mediterranean Sea is a bit risky. The fluxes of organic nutrients to the surface layer of the Mediterranean Sea are (probably) an overestimate. Your daily and average annual atmospheric fluxes are obtained from just to two points, off Marseille and in Lampedusa. Do you think that they are representative for the entire Mediterranean Sea? I do not believe that. Section 4.3 is very useful but you must prevent to give the impression that your calculations can be extrapolated to the entire Mediterranean.

MINOR DETAILS

Lines 224 and 226. You are referring to Figure 2, not Figure 1. Please, correct.

Line 422. These numbers should be divided by 1000 or expressed in mol C.

Line 680, Table 2. Please, add a column with the initial concentrations of DOC even if it is redundant.

---

## Author Comment (AC1) · 23 Sep 2020

Response to the 1st referee We would like to thank the referee for her relevant comments and suggestions on our submitted manuscript. Here below, we address our response (in bold and in italic) and we highlighted the text modifications in the revised manuscript (in italic between inverted commas). The pdf file is attached below.

Referee 1 1. The manuscript title is a bit too long, consider rewriting. Two suggestions: "Experimental evidence of the potential availability of organic matter from aerosols to heterotrophic bacteria" or "Potential bioavailability of organic matter from aerosols to heterotrophic bacteria".

We agree with the reviewer. The title of the manuscript was shortened in the revised manuscript following the suggestion of the referee.

Modified title (lines: 1-2): ‘Potential bioavailability of organic matter from atmospheric particles to marine heterotrophic bacteria’ 2. I would be careful when addressing DOC budgets, extensively mentioned in the introduction and discussion and comparing with budgets estimated from the experiments. The dust sources used in this experiment were artificially dissolved prior to incubation, but they arrive in the environment as POC. Is it realistic that, in the environment, dust particles would stay enough time at the ocean’s surface to be dissolved to that proportion found in Table 1 before sinking? This could be addressed better in the discussion. We may disagree with the referee in this point. In the case of wet deposition events, a substantial fraction of atmospheric particles components gets dissolved in rainwater before deposition in the ocean and, therefore, organic carbon arrives in the environment as a mixture of DOC and POC. In this context, by artificially leaching aerosols in ultrapure water prior to amendments, the experimental design applied in this study mimicked a wet deposition event and focused on water-soluble organic carbon. We have clarified this point in the Methods section and in the discussion as suggested by the referee.

Modified text (lines: 130-136): ‘To do so, particles from two whole Saharan dust filters and from 7.9 x 7.9 cm2 of the anthropogenic aerosol filter were firstly leached, each in 650 mL ultrapure water. After being sonicated for 40 min, the suspended particles were filtered through pre-combusted GF/F filters (450 °C, 6 h) to recover the dissolved fraction. A volume of 200 mL of each leachate was finally introduced in the corresponding aerosol amended treatments (D and A). Thus, this protocol simulated an input of atmospheric water-soluble organic carbon through aerosol wet deposition, i.e. an input of atmospheric DOC’.

Modified text (lines: 413-416):

‘Note that this assumption would apply to the case of wet deposition events in which an

important fraction of atmospheric particles is solubilized in rainwater before deposition. In the case of dry deposition events, the labile and refractory fractions would depend on the potential solubility of atmospheric organic matter in seawater and its residence time in the euphotic layer.'

3- Introduction: in Line 65: "Nutrient availability and microbial community structure regulate the accumulation and the remineralization of DOM, influencing export efficiency". I do not feel comfortable suggesting papers of mine, but we addressed this issue through incubation experiments with nutrient amendments and observed DOC and DON dynamics. Bibliography: Bif, M.B.; Hansell, D.A.; Popendorf, K.J. Controls on the fate of dissolved organic carbon under contrasting upwelling conditions. Frontiers in Marine Science, v.5 (463), 2018. I think this can contribute to both introduction and discussion sections. We would like to thank the referee for this suggestion. This publication was added in the introduction. Modified text (line 65): 'This percentage reaches more than 50% of the total carbon export in the oligotrophic oceans (Carlson et al., 1994; Guyennon et al., 2015; Letscher and Moore, 2015; Roschan and DeVeries, 2017; Bif et al., 2018).' In Lines 76-88: Although atmospheric dust is of undoubted importance in oligotrophic regions of the Atlantic and Mediterranean Sea, it does not play a role in other oligotrophic regions such as in the South Pacific Subtropical Gyre. This could be better addressed by adding one or two sentences as it gave the impression that dust deposition is important in every oligotrophic region. Example of bibliography: Jickells, T. D., et al. 2005. Global iron connections between desert dust, ocean biogeochemistry and climate, Science, 308, pp.67–71. We agree with the referee. This paragraph was clarified in the revised manuscript as follow: Modified text (Lines: 82-84) 'By bringing new nutrients to the upper waters, atmospheric deposition plays a key role in some oligotrophic regions such as the Mediterranean Sea and the Northern Atlantic and Pacific gyres, particularly under stratified conditions (Guieu et al., 2014; Letelier et al., 2019).' ' In the control and glucose treatment, why didn't the authors add Fe to the solution? This is a limiting nutrient for heterotrophic bacteria, is probably found in very high concentrations in the dust treatments and could make a difference in the C and G

incubations.

We understand the concern of the reviewer. However, dust may have provided not only Fe, but also other trace metals that could be potentially stimulating or toxic to bacteria. Given the difficulty to reconstitute artificially the mixed trace metal composition of dust, we decided not to amend the C and G incubations with metals. Please note, however, that even without iron addition, the utilization of carbon in the glucose treatment was higher than that observed in the dust treatment, suggesting that the availability of iron did not constrain carbon utilization in the glucose treatment. This might be explained by the lack of Fe limitation of heterotrophic bacteria in the Mediterranean Sea (Guieu et al., 2002). In addition, the inoculum may have certainly provided Fe to the C and G incubations. Although our experimental conditions were fully adapted for the analysis of nutrients and organic matter, they did not meet the requirements for trace metal analysis. Indeed, trace metal clean conditions would have been necessary to explore this possibility.

In the paragraph starting in l.122, was DOC analyzed using the method described in the paragraph starting in l.178? Please clarify.

Yes, DOC concentrations were analyzed following the procedure described in the section (2.3). In the revised manuscript, a sentence was added to refer the reader to the analytical procedure.

Modified text (line: 124):

'The DOC concentration in the artificial seawater was 6 $\mu$M (see section 2.3 for the analytical method).'

Line 178: Replace "online from" with "in line with". We corrected it in the revised manuscript. Modified text (Line 188): 'Samples for DOC analysis (10 mL) were filtered in line through pre-combusted (450 °C, 6 h) 47-mm GF/F filters.' In section 3.1. I see an overall DOC increase in the C treatment instead of a 5_M decrease, observed in

Figure 2, Panel A. The decrease of DOC was calculated by considering the observed maximal and minimal values in the control. This clarification was added in Table 2. In the control treatment, the decrease of DOC was calculated between T=5.7 d and Tfinal as an increase was observed at the beginning of the incubation experiment. Please see Table 2 in the revised manuscript.

Please also note the supplement to this comment:
https://bg.copernicus.org/preprints/bg-2020-187/bg-2020-187-AC1-supplement.pdf

---

## Author Comment (AC2) · 23 Sep 2020

Response to the 2nd referee We would like to thank the referee for his/her relevant comments and suggestions on our submitted manuscript. Here below, we address our response (in bold and in italic) and we highlighted the text modifications in the revised manuscript (in italic between inverted commas). Our response (pdf) is downloaded below as a supplement. Referee 2 1) Inorganic nutrients were added to the microbial cultures to avoid inorganic nutrient limitation. Final concentrations were 1 uM-N and 0.3 uM-P, which are well above the expected concentration in the surface layer of an oligotrophic region. Given that inorganic nutrients have been added in the same concentrations to all treatments, comparison among treatments is valid but: what about extrapolation of your results to oligotrophic conditions in the field? 2) The WSF of Sahara dust and anthropogenic aerosols do not contain only DON and DOP but also inorganic N (ammonium, nitrite and nitrate) and phosphate. Therefore, even if you have not added inorganic nutrients to the microbial cultures (see point 1), inorganic N and P would be added in the WSF. In this regard, it is relevant that the concentrations of inorganic N and P in the WSF are presented and discussed; We agree with the reviewer on the relevance of inorganic nutrient data. In the revised manuscript, initial inorganic nutrient concentrations were added in the method section and discussed (line: 325-330). Modified text in the method (Lines: 137-142) 'To avoid nutrient limitation, artificial seawater was enriched with nitrogen (NH4Cl + NaNO3) and phosphate (KH2PO4), to final concentrations of 1 $\mu$M and 0.3 $\mu$M in the incubation bottles, respectively. Therefore, initial nutrient concentrations in the control and glucose treatments were 1.02 $\pm$ 0.02 $\mu$M and 1.02 $\pm$ 0.05 $\mu$M for nitrogen, respectively and, 0.29 $\pm$ 0.01 $\mu$M and 0.27 $\pm$ 0.02 for phosphorus. However, in anthropogenic and dust treatments, aerosols amendments increased nutrient concentrations in the incubation bottles, reaching 6.02 $\pm$ 0.34 in Saharan dust treatment and 7.03 $\pm$ 0.09 in anthropogenic treatment for nitrogen concentration and, similarly reaching 0.40 $\pm$ 0.02 in Saharan dust treatment and 0.34 $\pm$ 0.01 in anthropogenic treatment for phosphorus concentration.' Modified text in the discussion (Lines: 325-330) 'Alongside to those nutrient enrichments, aerosol amendments resulted in initial N concentrations up to 7 $\mu$M (6 $\mu$M and 7 $\mu$M in D and A, respectively) and initial P concentrations up to 0.4 $\mu$M (0.40 $\mu$M and 0.34 in D and A, respectively). Different BGE values were observed between D- and A-treatments despite similar initial nutrient concentration. Moreover, BGE in D was similar to that of the G treatment despite a higher nutrient concentration, suggesting that inorganic nutrients were weakly involved in the control of BGE.'

We agree with the reviewer that the concentrations of nitrogen and phosphorus in the Mediterranean Sea are low in the upper water, especially during the stratification period during which nutrient concentrations fall into nanomolar levels. However, the added

concentrations were chosen to avoid nutrient limitation over the incubation period, particularly in the glucose treatment. Furthermore, in this study, the seawater used for the inoculum of heterotrophic bacteria was sampled during October. During that period, nitrogen and phosphorus concentrations increase within the surface waters as the mixed layer deepens. During the mixing period, nitrogen and phosphorus concentrations could reach up to 2 and 0.25 $\mu$M (Pasqueron de Fommervault et al., 2015), respectively in the upper water of the Mediterranean Sea. Those reported values are in the same order of magnitude as N and P concentrations added in the framework of this study. Of course, we agree that any extrapolation of experimental results to the field can be debated as the encountered variables in natural environments cannot all be reproduced in the context of experimental work. If low inorganic N and P concentrations are encountered in the upper waters, their bioavailability may constrain/ limit the utilization of atmospheric DOC. However, as expected by the reviewer, aerosol amendments delivered inorganic nitrogen and phosphorus , resulting in initial concentrations of 6.02 $\pm$ 0.34 in Saharan dust treatment and 7.03 $\pm$ 0.09 in anthropogenic treatment for nitrogen concentration and, similarly reaching 0.40 $\pm$ 0.02 in Saharan dust treatment and 0.34 $\pm$ 0.01 in anthropogenic treatment for phosphorus concentration. In the upper water of oligotrophic regions, N and P supply from atmospheric deposition could alleviate nutrient limitation, thus allowing the utilization of atmospheric DOC. In addition, although inorganic N and P could limit bacterial activity in such oligotrophic regions, DOP and DON concentrations are much higher in the upper waters than inorganic N and P and sustain microbial activity as shown in several nutrient depleted areas (i.e. Van Wambeke et al., 2002; 2020; Mather et al., 2008).

3) It is reported that the DOC concentration in artificial seawater was 6 uM and in the WSF <0.3 uM (once diluted in the ASW). However, the average DOC concentration in the control treatments was 19 uM (calculated from Table 2). What caused this difference? Should we assume that it also occurs in treatments G, A and D?; The concentration of DOC in the control treatment at T0 (T0=6 hours) was indeed higher than the one expected (6.3 $\mu$M). We don't have a straightforward explanation for this

observed difference. However, a potential release of carbon from heterotrophic bacteria could have occurred, following their senescence as organic matter was not provided in that treatment. This suggestion could be supported by the extremely low bacterial abundance and production observed for that treatment along the incubation experiment. It is unlikely that this also occurred in G, A and D-treatments since initial DOC concentration ($40 \pm 3$ $\mu$M, $36 \pm 2$ $\mu$M and $34 \pm 3$ $\mu$M) agrees with the amount of DOC added through the amendments (see Methods section, lines 130-135).

4) The estimates of LDOC(%) are obtained for the 4 treatments (C, G, A and D) comparing the initial DOC and the DOC decrease of each treatment. However, a considerable part of the initial DOC is already present in the control treatment (see point 3). Therefore, how should we interpret the LDOC(%) numbers in Table 2? For example, if the DOC decrease in the control treatment is 5 uM and in the G treatment is 22 uM, the DOC decrease exclusively due to the glucose addition should be 17 uM. Given that the DOC of the glucose addition is 40 uM and in the control is 19 uM, the LDOC(%)of the glucose addition would be 81% (=(22 − 5) / (40 − 19)). This is very different from the 55% in Table 2. For treatments A and D the difference is not so large but it is conceptually important; 5) The same reasoning is applicable to the BGE(%) calculations (LDOC is in the denominator of the formulae). For example, for the G treatment BGE(%) should be 9% (=(1.7 − 0.7)/ (22 − 5) and for the D treatment 26% (=(1.19 − 0.17)/(9 − 5)). It really makes a difference; We understand the Reviewer's reasoning. However, as mentioned previously, if the 'extra' DOC concentration observed in the control treatment had occurred in the glucose treatment (as well as in anthropogenic and Saharan dust treatment) we would have observed an additional 19 $\mu$M at T0 for each treatment compared to the actual DOC added, which was not the case. Therefore, we do not think that taking into account the decrease of DOC observed in the control for the calculation of the LDOC and BGE in glucose, Saharan dust and anthropogenic treatments is the appropriate approach. Indeed, if 5 $\mu$M decrease of DOC in the control treatment resulted in an initial concentration of 19 $\mu$M in, it is unlikely that the background of labile DOC, if there is, was the same in A, D and G treatments as

any 'extra' DOC was observed in those treatments.

6) Also concerning BGE (%), it should be better to use the bacterial biomass, calculated from BA with a conversion factor, rather than BP; and We recognize that the use of BP in the calculation of BGE could be biased as BP is determined using a leucine to carbon conversion factor. Nevertheless, the use of BA for BGE calculation could be biased as well as: i) it requires the utilization of abundance/biomass conversion factor and biovolume varies substantially during growth. ii) The variation of bacterial abundance is a net variation, reflecting a balance between growth and mortality. This is visible through observations of apparent growth rates calculated from the exponential growth phases of BA and BP, which are lower for abundances than for BP. It is also visible when we look to the lag phase, which was shorter for BP than for BA (Table 3), as BA is the sum of active and inactive cells. This is a classical problem occurring during dilution experiments, used to determine BGEs but also leucine-conversion factors (Kirchman et al., 1982; Amermann et al., 1984). For these reasons we prefer to use BP data for BGE calculations.

Ammermann, J. W., Furhman, J. A., Hagström, A., and Azam, F.: Bacterioplankton growth in sea water : I. Growth kinetics and cellular characteristics in sea water cultures, Mar. Ecol. Prog. Ser., 18, 31-39, 1984. Kirchman, D. L., Ducklow, H. W., and Mitchell, R.: Estimates of bacterial growth from changes in uptake rates and biomass., Appl. Environ. Microbiol., 44, 1296-1307, 1982.

7) Extrapolation of your results to the entire Mediterranean Sea is a bit risky. The fluxes of organic nutrients to the surface layer of the Mediterranean Sea are (probably) an overestimate. Your daily and average annual atmospheric fluxes are obtained from just to two points, off Marseille and in Lampedusa. Do you think that they are representative for the entire Mediterranean Sea? I do not believe that. Section 4.3 is very useful but you must prevent to give the impression that your calculations can be extrapolated to the entire Mediterranean. We fully agree with the reviewer that section 4.3 is very useful but risky. It was not our intention to extrapolate our results to the whole

Mediterranean basin. The main assumption under the proposed possible influence of our results in marine C cycle in the Mediterranean Sea was that the labile/refractory partitioning obtained in our study is valid for reported atmospheric fluxes of DOC. That is why we keep the estimated fluxes in this section per square meter, in order not to extrapolate to the entire basin. In the revised version of the manuscript, we have rewritten the first part of the section in order to qualify the ideas. Modified text (lines: 403-408): 'Atmospheric fluxes of DOC in the Mediterranean Sea have been reported to be 6 times higher than river fluxes (Djaoudi et al., 2018; Galleti et al., 2020 this is-sue), highlighting atmospheric deposition as a major allochthonous source of carbon to this marine region. Hereafter, we apply the partitioning between labile (LDOC) and refractory DOC (RDOC) in aerosols obtained in this study to these previously reported atmospheric DOC fluxes in the Mediterranean Sea to propose a tentative estimation of LDOC and RDOC fluxes from both Saharan dust and anthropogenic deposition and to which extent they could potentially contribute to marine carbon cycle'.

MINOR DETAILS Lines 224 and 226. You are referring to Figure 2, not Figure 1. Please, correct. Thank you for this comment. This mistake was corrected in the revised manuscript.

Modified text (Lines: 234-236) 'The initial DON and DOP concentrations in the C-treatment were $1.6 \pm 0.1$ $\mu$M and below the detection limit, respectively. The G-treatment exhibited the same initial DON concentration as the C-treatment, with values of $1.8 \pm 0.2$ $\mu$M while DOP concentrations reached $0.03 \pm 0.01$ (Fig. 2B, C). In A and D-treatments, the concentration of both DON and DOP increased immediately after aerosol addition (T0). The DON concentration reached $4.3 \pm 0.7$ $\mu$M and $3.7 \pm 0.2$ $\mu$M in the A and D-treatments, respectively (Fig. 2A), resulting in lower C:N and higher N:P elemental ratios in A and D treatments than in G-treatment (Table S2).'

Line 422. These numbers should be divided by 1000 or expressed in mol C. Yes, we agree, and we apologize for this mistake. We corrected this paragraph as follow:

Modified text (Lines: 336-441): 'By applying the percentages of LDOC observed in this to previously daily reported atmospheric fluxes (0.03-1.78 mmol C m-2 day-1) (Djaoudi et al., 2018; Galleti et al., 2020 this special issue), daily atmospheric fluxes of LDOC would range between 0.007-0.534 mmol C m-2 day-1 for Saharan dust deposition and between 0.006-0.480 mmol C m-2 day-1 for anthropogenic deposition. These fluxes would thus represent up to 13% of photosynthetic DOC production, supporting the hypothesis of a potential role of atmospheric deposition of DOC in sustaining secondary production.'

Line 680, Table 2. Please, add a column with the initial concentrations of DOC even if it is redundant. We added this column in the revised manuscript. Please see table. 2 attached below as a figure (please see figure 1 in the caption below).

Please also note the supplement to this comment:
https://bg.copernicus.org/preprints/bg-2020-187/bg-2020-187-AC2-supplement.pdf

**Table 2. Initial dissolved organic carbon concentration and its decrease and contribution to the initial DOC stocks (expressed as a percentage of labile dissolved organic carbon, LDOC).**

|  | Initial DOC; μmol C L-1 | DOC decrease; μmol C L-1 | LDOC [%] |
|---|---|---|---|
| C1 | 22 | 5 | 23 |
| C2 | 22 | 6 | 27 |
| C3 | 18 | 4 | 22 |
| **Mean (± SD)** | **20 ± 2** | **5 ± 1** | **24 ± 3** |
| G1 | 36 | 19 | 53 |
| G2 | 42 | 24 | 57 |
| G3 | 41 | 22.5 | 54 |
| **Mean (± SD)** | **40 ± 3** | **22 ± 3** | **55 ± 2** |
| A1 | 34 | 6 | 17 |
| A2 | 37 | 11 | 29 |
| **Mean (± SD)** | **36** | **9 ± 4** | **24 ± 8** |
| D1 | 31 | 7 | 22 |
| D2 | 36 | 8.5 | 24 |
| D3 | 36 | 11 | 30 |
| **Mean (± SD)** | **34 ± 3** | **9 ± 2** | **25 ± 4** |

**Fig. 1.**

---

## Author Response (AR2)

We would like to thank the editor in chief for her relevant suggestion. This was addressed in the result and the discussion sections. All text modifications are highlighted below in italic, between commas.

Editor in chief

I would like to thank the authors who responded to most of the comments and suggestions by making relevant additions to the text.
In particular, all responses to reviewer 1 have been nicely addressed.
For reviewer 2, although the authors gave detailed and very well argued (and therefore convincing) responses, they did not propose additional text in the manuscript to some comments. However, I believe that their argumentation is also necessary in the revised version (even by adding only a short summary) for readers that could have the same remarks as Rev1.

This concerns: the difference between expected and measured DOC concentration in control and the possible consequences on the calculations made in the different treatments for LDOC and BGE; the choice of using BP rather than BA to calculate BGE (an interesting point); most of the arguments developed in your response could be added to the Results section.
I would encourage the authors to provide those short additions before the paper can be accepted for publication.
Thank you

**The difference between expected and measured DOC concentration in the control treatment, and consequences on the calculation of LDOC were added in the result section (lines: 226-236).**

Modified text (lines: 225-235)

'*One hour after aerosols amendments ($T_0$), the measured initial concentration of DOC in the control treatment ($14 \pm 1$ $\mu M$) was higher than the expected value of 6.3 $\mu M$ (DOC concentration in artificial seawater + inoculum, see method section). This 'extra DOC' was not observed in the amended treatments, where concentrations of $40 \pm 3$ $\mu M$ 36 $\mu M$ and $34 \pm 3$ $\mu M$ were observed in G, A and D treatments, respectively, consistent with the amount of DOC added through amendments (36 $\mu M$ final concentration in the incubation bottles, see method section) (Table 1). Over the incubation period ($T_0$-$T_{final}$), DOC concentration decreased in the three amended treatments (Fig. 2A). This decrease was highest in the G-treatment ($22 \pm 2$ $\mu M$) followed by both D ($9 \pm 2$ $\mu M$) and A ($9$ $\mu M$) treatments (Table 2; Fig. 2A). In the C-treatment, a net decrease of DOC, of $5 \pm 1$ $\mu M$, was detected only after $T_{5.7}$ (between $T_{5.7}$ and $T_{final}$). As no 'extra DOC' was observed in G, A and D treatments, the variation of DOC in the control was not taken into consideration in the calculation of labile DOC in G, D and A treatments. Therefore, the resulted labile DOC fractions were $55 \pm 2\%$, $25 \pm 4\%$ and $24 \pm 8\%$ in G, D and A treatments, respectively (Table 2).*'

**The use of BP rather than BA in the calculation of BGE is addressed in the discussion section (lines: 323-331). Growth rates calculated using BA were also added in the table 3.**

Modified text (lines: 323-330)

'*The use of BP in the calculation of BGE could be biased as BP is determined using a leucine to carbon conversion factor. However, in this study, we still preferred using BP rather than BA in this calculation, as BA could be also subjected to a number of biases. Indeed, the determination of BGE using BA requires the utilization of abundance/biomass conversion factor and, yet biovolume varies substantially during the growth. Furthermore, the variation of cell abundance is the result of a net balance between growth and mortality. This is visible through observations of apparent growth rates, calculated from exponential growth phases of both BP and BA (Table 3), lower for BA than BP. Likewise, this is also visible regarding the lag phase, shorter for BP than BA (Table 3). This is a classical issue occurring during dilution experiments for BGEs but also leucine-conversion factors determinations (Kirchman et al., 1982; Ammerman et al., 1984).*'

Modified table 3

**Table 3. Duration of the exponential growth phase for bacterial abundance (BA) and bacterial production (BP), bacterial growth rate (µ) estimated from both BP and BA changes during the exponential growth phase, time integrated BP during the exponential growth phase and bacterial growth efficiency (BGE). Data from each triplicate and average (± SD) values are given for the control (C), glucose, anthropogenic and Saharan dust treatments.**

| Treatments | Exponential phase period for BA [days] | Exponential phase period for BP [days] | $\mu$; [d$^{-1}$] (estimated from BP) | $\mu$; [d$^{-1}$] (estimated from BA) | Time integrated BP$^{(b)}$ [µmol C L$^{-1}$ dt] | BGE $_{BP}$ [%] |
|---|---|---|---|---|---|---|
| C1 | 7.7 – 13.7 | 1.7 - 8.7 | 1.23 ± 0.19 | 1.41 ± 0.56 | 0.23 | 5 |
| C2 | 6.7 - 8.7 | 1.7 - 8.7 | 0.94 ± 0.15 | 0.86 ± 0.4 | 0.11 | 2 |
| C3 | 6.7 – 8.7 | 1.7 - 8.7 | 1 ± 0.13 | 0.94 ± 0.55 | 0.18 | 4.5 |
| **Mean (± SD)** | | | **1.05 ± 0.15** | **1.07 ± 0.39** | **0.17 ± 0.06** | **4 ± 2** |
| G1 | 3.7 - 5.7 | 0 – 5.7 | 1.99 ± 0.32 | 1.21 ± 0.08 | 1.07 | 5.6 |
| G2 | 2.7 – 4.7 | 0.7 – 3.7 | 3.44 ± 0.52 | 1.25 ± 0.49 | 1.87 | 7.8 |
| G3 | 2.7 – 3.7 | 0 – 3.7 | 3 ± 0.41 | 1.3 ± 0.66 | 2.16 | 9.6 |

| | | | | | | |
|---|---|---|---|---|---|---|
| **Mean (± SD)** | | | **2.81 ± 0.74** | **1.25 ± 0.05** | **1.70 ± 0.57** | **7.6 ± 2** |
| A1 | 5.7 – 8.7 | 2.7 – 6.7 | 1.64 ± 0.21 | 1.14 ± 0.07 | 0.11 | 1.8 |
| A2 | 4.7 – 8.7 | 2.7 – 6.7 | 1.62 ± 0.28 | 1.37 ± 0.15 | 0.19 | 1.7 |
| **Mean (± SD)** | | | **1.63** | **1.25** | **0.15** | **1.7** |
| D1 | 5.7 – 7.7 | 2.7 – 7.7 | 1.74 ± 0.09 | 1.05 ± 0.19 | 1.11 | 16.1 |
| D2 | 5.7 – 7.7 | 0.7 – 7.7 | 1.41 ± 0.10 | 1.01 ± 0.16 | 1.58 | 18.5 |
| D3 | 4.7 – 5.7 | 1.7 - 5.7 | 1.80 ± 0.30 | 1.10 ± 0.49 | 0.88 | 8.1 |
| **Mean (± SD)** | | | **1.65± 0.20** | **1.05± 0.04** | **1.19 ± 0.36** | **14.2 ± 5.5** |